# The neuron-specific IIS/FOXO transcriptome in aged animals reveals regulatory mechanisms of cognitive aging

Yifei Weng[1], Shiyi Zhou[1], Katherine Morillo[1], Rachel Kaletsky[1,2], Sarah Lin[1], Coleen T Murphy[1,2]*

[1]Department of Molecular Biology, Princeton University, Princeton, United States; [2]Princeton University, Princeton, United States

*For correspondence: ctmurphy@princeton.edu

Competing interest: The authors declare that no competing interests exist.

**Abstract** Cognitive decline is a significant health concern in our aging society. Here, we used the model organism *C. elegans* to investigate the impact of the IIS/FOXO pathway on age-related cognitive decline. The *daf-2* Insulin/IGF-1 receptor mutant exhibits a significant extension of learning and memory span with age compared to wild-type worms, an effect that is dependent on the DAF-16 transcription factor. To identify possible mechanisms by which aging *daf-2* mutants maintain learning and memory with age while wild-type worms lose neuronal function, we carried out neuron-specific transcriptomic analysis in aged animals. We observed downregulation of neuronal genes and upregulation of transcriptional regulation genes in aging wild-type neurons. By contrast, IIS/FOXO pathway mutants exhibit distinct neuronal transcriptomic alterations in response to cognitive aging, including upregulation of stress response genes and downregulation of specific insulin signaling genes. We tested the roles of significantly transcriptionally-changed genes in regulating cognitive functions, identifying novel regulators of learning and memory. In addition to other mechanistic insights, a comparison of the aged vs young *daf-2* neuronal transcriptome revealed that a new set of potentially neuroprotective genes is upregulated; instead of simply mimicking a young state, *daf-2* may enhance neuronal resilience to accumulation of harm and take a more active approach to combat aging. These findings suggest a potential mechanism for regulating cognitive function with age and offer insights into novel therapeutic targets for age-related cognitive decline.

## eLife assessment

This **fundamental** study investigates the transcriptional changes in neurons that underlie loss of learning and memory with age in *C. elegans*, and how cognition is maintained in insulin/IGF-1-like signaling mutants. The presented evidence is **compelling**, utilizing a cutting-edge method to isolate neurons from worms for genomics that is clearly conveyed with a rigorous experimental approach. Overall, this study supports that older daf-2 worms maintain cognitive function via mechanisms that are unique from younger wild type worms, which will be of great interest to neuroscientists and researchers studying ageing.

## Introduction

The loss of cognitive function is a rising problem in our aging society. A 2008 study estimated that at least 22.2% (about 5.4 million) of individuals over the age of 71 in the United States have at least mild cognitive impairment (*Plassman, 2008*; *Langa and Levine, 2014*; *Gillis et al., 2019*). Furthermore, global dementia cases are predicted to triple from an estimated 57.4 million cases in 2019–152.8 million cases in 2050 (*Feigin et al., 2020*; *Nichols and Vos, 2020*). As most industrialized

countries are experiencing a rapid increase in the proportion of the aged population, understanding and potentially preventing the underlying issues of neuronal structural and behavioral decline associated with aging is crucial for societal health.

*C. elegans* is an excellent model system for studying neuronal aging, given its tractable genetics, short lifespan, and simple nervous system (*White et al., 1986*). Most importantly, *C. elegans* experiences rapid loss of learning and memory with age (*Kauffman et al., 2010*): by Day 4 of adulthood, all long-term associative memory ability is lost, and by Day 8, *C. elegans* cannot carry out associative learning or short-term associative memory (*Kauffman et al., 2010*) – despite the fact that these worms can still move and chemotaxis perfectly well. That is, with age worms first lose long-term memory ability (by Day 4), then short-term memory and learning ability (Day 6–8), then chemotaxis (Day 10–12), then motility (Day 16) (*Kauffman et al., 2010*; *Hahm et al., 2015*). Because learning and memory decline extremely early, we consider worms that are only a week old to already be 'cognitively aged,' despite the fact that they can chemotaxis and move well, and will continue to live for another one to two weeks. Therefore, we can examine neurons from these 7–8-day-old adults to explore the causes of these cognitive declines in animals that are otherwise quite healthy. Many human neuronal aging phenotypes and genes of interest for mammalian neuronal function are conserved in *C. elegans* (*Arey and Murphy, 2017*), making discoveries in *C. elegans* possibly applicable to humans.

The Insulin/IGF-1-like signaling (IIS)/FOXO pathway was first discovered to play a role in longevity in *C. elegans*. The lifespan of *daf-2*/Insulin/IGF-1 receptor mutants is twice that of wild-type animals (*Kenyon et al., 1993*), and this lifespan extension requires the downstream Forkhead box O (FOXO) transcription factor DAF-16 (*Kenyon et al., 1993*). DAF-16/FOXO controls the expression of many genes that contribute to longevity, including stress response, proteostasis, autophagy, antimicrobial, and metabolic genes (*Murphy et al., 2003*). As a conserved regulator, the IIS/FOXO pathway also regulates longevity in *Drosophila*, mice, and humans (*Clancy et al., 2001*; *Blüher et al., 2003*; *Suh et al., 2008*; *Willcox et al., 2008*). In addition to regulating lifespan, the IIS pathway regulates neuronal function via the FOXO transcription factor. In particular, *C. elegans* IIS/*daf-2* mutants display DAF-16-dependent improved learning, short-term memory, and long-term memory (*Kauffman et al., 2010*). While both young and old *daf-2* adult worms display increased learning and memory relative to wild-type, the duration of this extension is not known, and the mechanisms by which *daf-2* mutants maintain neuronal function in older worms are not yet understood. Compared to wild-type worms, *daf-2* mutants better maintain maximum velocity (*Hahm et al., 2015*), motility (*Liu et al., 2013*; *Li et al., 2016*), neuromuscular junctions, the ability to regenerate axons (*Byrne et al., 2014*; *Lakhina et al., 2019*), and neuronal morphology with age (*Pan et al., 2011*; *Tank et al., 2011*; *Toth et al., 2012*). In particular, we previously showed that while *daf-2* has lower observed motility on food (*Bansal et al., 2015*), this apparent is due to its high levels of a food receptor, ODR-10 (*Hahm et al., 2015*), and its downregulation reveals the much higher mobility of *daf-2* animals (*Hahm et al., 2015*), even on food, in addition to its much higher and maintained maximum velocity with age. Previously, we found that *daf-2* worms also extend learning beyond the wild-type's ability (*Kauffman et al., 2010*), but the full duration of this extension with age was not known. That is, exactly how late in life *daf-2* mutants can still learn and remember, and whether this is proportional to their lifespan extension, was not previously determined.

We previously performed neuron-specific RNA-sequencing in young (Day 1) adult *C. elegans* and identified neuron-specific targets (*Kaletsky et al., 2016*); genes upregulated in *daf-2* mutant neurons are distinct from those in the whole animal, and we found that these neuronal genes are necessary for the observed improvements in memory and axon regeneration in *daf-2* mutant worms. However, whether *daf-2* uses the same or different genes in young and old worms to improve and maintain cognitive function with age is unknown. Recent datasets using whole-animal single-cell RNA-seq have been generated for wild-type and *daf-2* worms, and these are sufficient for whole-body aging and pseudobulk analyses (*Gao et al., 2023*; *Wang et al., 2022*; *Roux et al., 2023*), but we have found that those data are not deep enough to use specifically for in-depth analysis of neurons, which can be difficult to gather from whole animals. Other data are from larval stages and cannot be extrapolated to aging adults (*Taylor et al., 2021*). To identify the transcriptional differences in the aging nervous system that might contribute to the loss of neuronal function with age in wild-type worms and the differences responsible for the extended abilities of *daf-2* animals, here we performed RNA sequencing on FACS-isolated neurons of aged (Day 8) wild-type and IIS/FOXO mutants. To further investigate

the role of the neuronal IIS/FOXO pathway, we identified genes both upregulated by the IIS/FOXO pathway, and genes that are differentially expressed in *daf-2* mutants with age. We found that *daf-2* differentially-regulated genes in the aged neurons are different from young neurons; in fact, many of these Day 8 *daf-2* vs *daf-16;daf-2* upregulated genes are stress response and proteolysis genes that may promote neuronal function and health. We then used functional assays to assess the contributions of *daf-2*-regulated genes to learning and memory. Our results suggest that *daf-2*'s neuronal targets in older worms are required to maintain neuronal functions with age, suggesting that additional and alternative mechanisms are at work in these aged mutants from their young counterparts.

## Results

### Wild-type neurons lose their neuronal function and identity with age

Previously, we found that cognitive abilities in *C. elegans*, including learning, short-term memory, and long-term memory, all decline with age (*Kauffman et al., 2010*). Moreover, neuronal morphology and regeneration ability are also impaired with age (*Byrne et al., 2014*; *Pan et al., 2011*; *Tank et al., 2011*; *Toth et al., 2012*). However, how these phenotypes are regulated at the molecular level in aging neurons remains to be systematically characterized. Therefore, we were interested in first identifying gene expression changes with age in wild-type neurons to characterize the normal physiological aging process. Before choosing timepoints to assess neuronal transcriptome changes, we carried out associative learning and short-term associative memory assays (*Kauffman et al., 2010*) as we have previously described (*Kauffman et al., 2010*; *Kauffman et al., 2011*; *Stein and Murphy, 2012*; *Stein and Murphy, 2014*). Briefly, well-fed worms are starved for 1 hr, then re-fed while exposed to the neutral odorant butanone for 1 hr; a choice assay between butanone and control immediately after training tests associative learning, while a choice assay after 1 hr of recovery on food-only plates tests short-term associative memory (*Kauffman et al., 2010*). Adult Day 1 worms are fully developed, young, and healthy, while wild-type Day 7–8 worms, although still in their mid-life, have completely lost their learning and short-term memory abilities already by Day 7 (*Kauffman et al., 2010*; *Figure 1a*), thus we consider them 'aged' for the purposes of understanding loss of cognitive ability. Therefore, we reasoned that a comparison of adult wild-type Day 1 neurons with wild-type neurons that are at least aged Day 7–8 should reveal changes with age that result in loss of cognitive function.

To identify genes that regulate age-related morphological and functional decline in wild-type neurons, we performed neuron-specific transcriptomic analysis using our previous FACS neuronal isolation method (*Kaletsky et al., 2016*) on six biological replicates each of Day 1 and Day 8 adult wild-type worms, where 100,000 GFP + cells were collected for each sample (*Figure 1—figure supplement 1a–c*). Because we previously found that whole-worm analyses mask changes found specifically in neurons (*Kaletsky et al., 2016*), to complement our aging neuron studies, we also carried out RNA-sequencing analyses of aging whole worms (*Figure 1—figure supplement 2a–e*), which we found is dominated by changes in the extracellular matrix (*Figure 1—figure supplement 2b*), stress response/pathogen genes (*Figure 1—figure supplement 2c*) and the alimentary system (intestine) (*Figure 1—figure supplement 2d*), overshadowing neuronal changes.

Principal components analysis of the FACS-isolated neuron RNA-seq samples indicated that they are well separated by age (*Figure 1b*), and downsampling analysis (*Robinson and Storey, 2014*; *Figure 1—figure supplement 1e*) suggested that we have sequenced to saturation, with an average of 41,636,463 uniquely-counted reads and detected the expression of 19725 coding and non-coding genes ($\log_{10}$(TPM) >0.5) (*Figure 1—figure supplement 1d*). Enrichment analysis of genes that are differentially expressed with age (*Figure 1c–e*) suggested that neuronal sorting and sequencing were successful because the sequenced genes are enriched for neuronal genes such as *mec-7*, *mec-12*, and *twk-49*, as expected, and less enriched for all other major tissues. Tissue enrichment analysis of differentially-expressed genes suggested that aging neurons lose genes most expressed in the nervous system and neurons (*Figure 1d*), as one might expect. Gene ontology (GO) analysis suggested that genes declining with age in neurons encode proteins important in neuronal function (*Figure 1e and f*), including synaptic proteins (e.g. *srh-59*, *rab-3*, *sng-1*, *sup-1*), potassium channels (e.g. *egl-23*, *twk-7*, *twk-49*, *ncs-5*), and transmembrane transporters (e.g. *folt-2*, *ccb-2*, *unc-79*, *exp-1*). The decrease in expression of these genes during aging may indicate that neurons are losing their identity and their

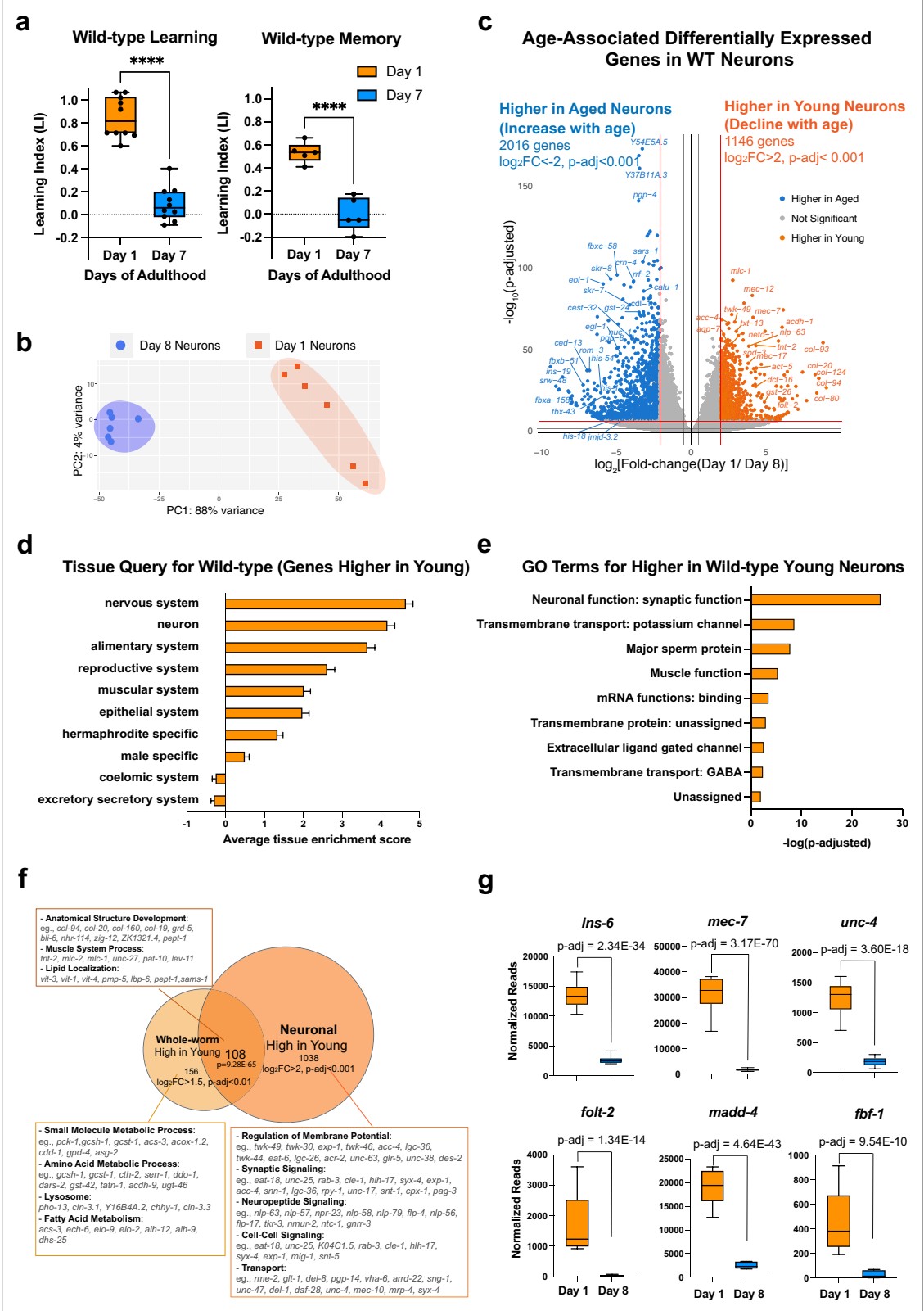

**Figure 1.** Identifying neuronal aging targets in wild-type (WT) worms using neuron-specific RNA-sequencing. (**a**) Wild-type learning and 1 hr memory results on Day 1 and Day 7. Learning and memory results are represented as learning index (LI). Details of the LI calculation are explained in the methods. Learning, n=10, memory, n=5. ****p<0.01. Student's t-test. (**b**) PCA plot for Day 1 (orange) and Day 8 (blue) neuronal bulk RNA-seq samples. (**c**) Volcano plot comparing age-associated differentially-expressed genes in WT neurons. Genes downregulated with age (orange) and upregulated

*Figure 1 continued*

with age (blue) were obtained by neuron-specific RNA sequencing of adult wild-type animals with neuron-specific GFP expression. Adjusted p-value <0.001, $\log_2$(Fold-change) >2. n=6 biological replicates per age. 1146 genes were significantly downregulated with age (higher in young neurons) and 2016 genes were upregulated with age (higher in old neurons) (**d**) Tissue prediction scores for genes higher in young neurons. (**e**) Gene ontology (GO) terms of genes that decline with age in wild-type neurons. Synaptic and signaling GO terms are enriched in neuronal genes. p-value calculated using hypergeometric distribution probability. (**f**) Comparison of whole-body higher-in-young genes and neuronal higher-in-young genes. GO Terms and representative genes were performed using g:Profiler software. P-value of overlapping regions were calculated using a hypergeometric calculator. (**g**) Normalized reads of *ins-6, unc-4, mec-7, folt-2, fbf-1,* and *madd-4,* in Day 1 and Day 8 neurons in our dataset. p-adjusted values were calculated from DESeq2 software. Box plots: center line, median; box range, 25-75th percentiles; whiskers denote minimum-maximum values.

The online version of this article includes the following figure supplement(s) for figure 1:

**Figure supplement 1.** Aged neuron-specific sequencing.

**Figure supplement 2.** Whole-worm RNA-sequencing identifies whole-body changes during aging.

ability to perform neuronal functions, such as signal transduction and axonal transport, and correlates with the behavioral and morphological declines observed in aging wild-type worms.

Comparing whole-worm sequencing and neuron-specific sequencing (*Figure 1f*), we found that genes involved in metabolic processes decline with age only in the body, and genes encoding structural proteins, lipid localization, and muscle system processes decline with age in both the body and in neurons, while neurons specifically lose genes that are associated with neuronal function, including synaptic proteins, neuropeptide signaling, and other neuron functions, correlating with neuronal loss of function with age. Together, these results indicate that neurons harbor many unique age-related changes that could be overshadowed in the whole-worm transcriptome but are revealed by neuron-specific sequencing.

Many genes that are more highly expressed in young neurons are known to be specific to a subset of neurons. *ins-6*, an insulin-like peptide specific to the ASI, ASJ, and AWA neurons (*Taylor et al., 2021*) that regulates longevity (*Artan et al., 2016*) and aversive learning (*Chen et al., 2013*), is significantly downregulated with age (*Figure 1g*). *srd-23*, a serpentine receptor located at the AWB neuron cilia (*Brear et al., 2014*), also decreases expression with age (*Figure 1—figure supplement 1f*). Furthermore, various genes specific to sensory neurons (*txt-12, flp-33*), touch neurons (*mec-7*), and motor neurons (*unc-4*) decline in expression with age (*Figure 1g, Figure 1—figure supplement 1f*). Previous studies showed that loss of genes including *ins-6* (*Chen et al., 2013*), *mec-7* (*Savage et al., 1989*), *unc-4, folt-2* (*Lakhina et al., 2019*), *madd-4* (*Maro et al., 2015*), and *fbf-1* (*Stein and Murphy, 2014*) lead to behavioral dysfunction in motility and chemosensory abilities; therefore, the decreased expression of these neuron type-specific genes with age may impact the function of individual neurons and disrupt neural circuit communication, ultimately contributing to the declines in behavior observed during aging (*Figure 1g*).

As neurons age, genes that increase in expression, while assigned to the nervous system (*Figure 2a*) are not specific for neuron function; instead, aged wild-type neurons express higher levels of many predicted F-box genes with predicted proteasome E3 activity (e.g. F-box proteins *fbxa-158, fbxb-51, pes-2.1,* and SKp1-related proteins *skr-12, skr-6*). Some transcription regulation (e.g. *ced-13, tbx-43, nhr-221, end-1*), and chromatin structure and function (e.g. *his-54, dot-1.2, jmjd-3.2, hil-7, utx-1*) genes also increase with age (*Figure 2b*), even though neurons appear to lose their neuron-specific transcriptional identity with age.

One ongoing discussion about changes during aging is how to interpret an increase in expression with age. There are two main models for genes that increase their expression with age and have a resulting impact on function: that they rise with age to compensate for lost function ('compensatory') and, therefore, promote function, or that their expression is deleterious to function and only rises with age through dysregulation. If a gene is compensatory, then its knockdown would abrogate learning and memory, even in young animals. If a gene's function is harmful to neurons, reducing its expression might be beneficial to the worm, even in young animals (Of course, there may be other scenarios in which a gene with multiple functions may be detrimental for some behaviors but beneficial for other physiological roles). To test this hypothesis, we reduced the expression of a small set of highly upregulated candidate genes in categories that might function in a compensatory manner. These include *utx-1*, a histone demethylase known to play a role in development (*Vandamme et al., 2012*) and lifespan in worms (*Jin et al., 2011; Maures et al., 2011; Guillermo et al., 2021*), and whose

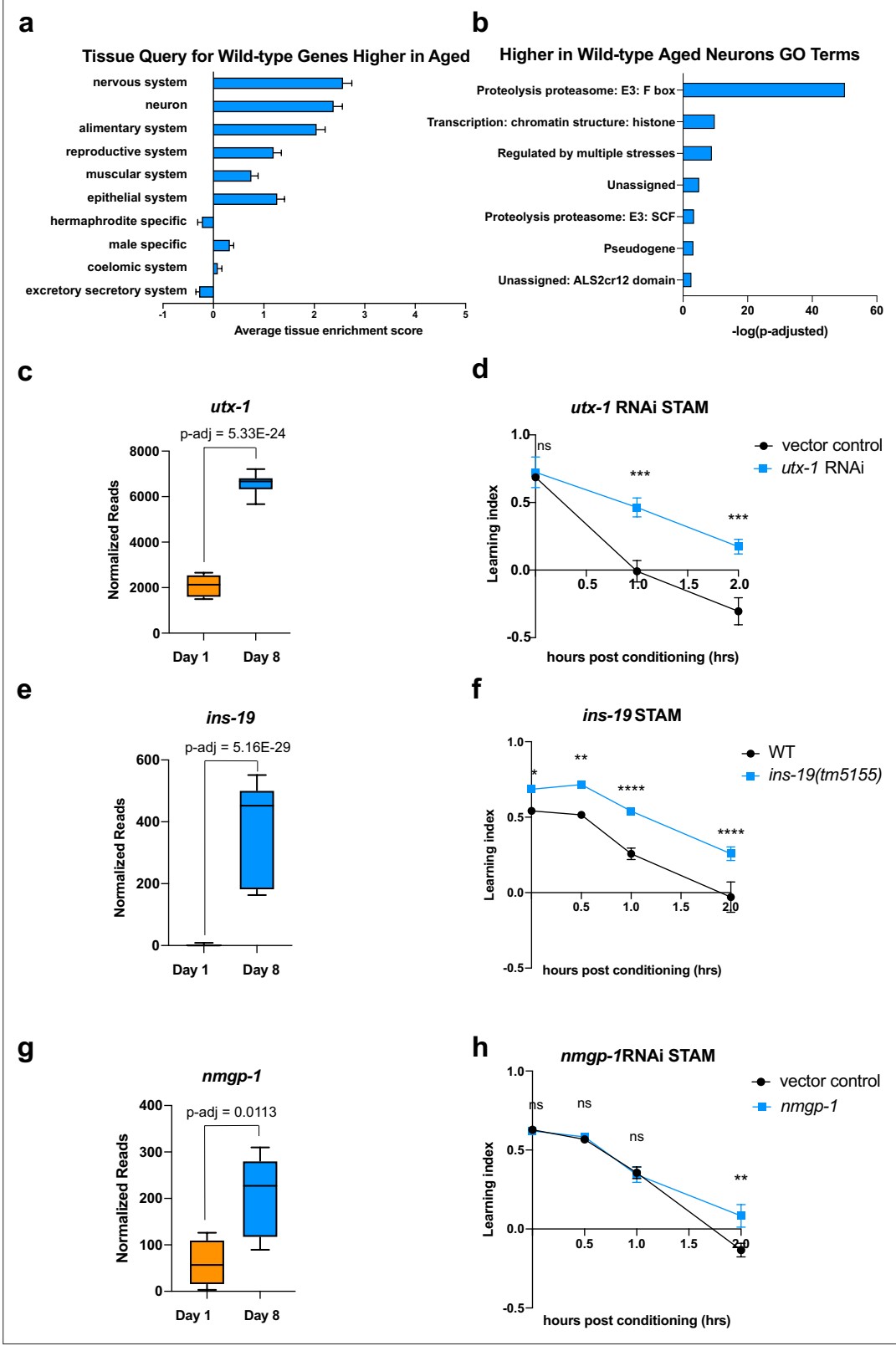

**Figure 2.** Genes that increase with age cause behavioral defects. (**a**) Tissue prediction score for wild-type genes expressed at higher levels in aged worms. (**b**) Gene ontology (GO) terms of genes expressed higher in aged neurons highlight transcription regulation and proteolysis. GO term analysis was done using Wormcat 2.0. (**c**) Normalized reads of *utx-1* on Day 1 and Day 8. (**d**) Short-term associative memory (STAM) assay shows that

*Figure 2 continued on next page*

*Figure 2 continued*

neuron-sensitized adult-only *utx-1* knockdown improves 1 hr and 2 hr memory of wild-type worms on Day 2. RNAi was performed using the neuron-RNAi sensitized strain LC108. (**e**) Normalized reads of *ins-19* on Day 1 and Day 8. (**f**) *ins-19* mutation improves learning and memory in STAM on Day 3 of adulthood. (**g**) Normalized reads of *nmgp-1* on Day 1 and Day 8. (**h**) *nmgp-1* neuron-sensitized RNAi knockdown improves memory in STAM on Day 2. RNAi was performed using the neuron-RNAi sensitized strain LC108.P-adj value of normalized count change generated from DEseq2 analysis. (**c, e, g**) Box plots: center line, median; box range, 25-75$^{th}$ percentiles; whiskers denote minimum-maximum values. Normalized reads and adjusted p-value were calculated using the DESeq2 software. Each dot represents one sequencing replicate. (**d, f, h**) n=5 plates in each behavioral experiment. Representative result of two biological repeats is shown. *p<0.05. **p<0.01. ***p<0.001. ****p<0.0001. Two-way ANOVA with Tukey's post-hoc analysis.

homolog has been implicated in cognition in mammals (*Shaw et al., 2023*; *Tang et al., 2017*); *ins-19*, an insulin-like peptide; and *nmgp-1*, a neuronal glycoprotein involved in chemosensation (*Fernández et al., 2022*). In each case, we see that gene expression is significantly higher in old than in young neurons (*Figure 2c, e and g*). If a gene increases expression to benefit neurons, we would expect to see no difference in memory in young animals where there is no defect; by contrast, if the increase of a gene is deleterious, we would expect to see an improvement in behavior when knocked down, even in young animals. We performed adult-only neuron-sensitized RNAi knockdown to prevent any possible deleterious effects caused by changes during development, which largely takes place in early larval stages; testing in young adult animals is logical because there is no memory in aged wild-type, so any deleterious effect of knocking down a potentially compensatory gene in an aged would not result in a change. For all behavioral assays, we first prioritized significantly-changed genes with high fold-change, and then those with mammalian homologs.

We found that 48 hr (L4-Day 2) of adult-only knockdown of *utx-1* increases 1 hr and 2 hr memory (*Figure 2d*), the loss-of-function mutation of *ins-19* increases both learning and memory (*Figure 2f*) and the adult-only knock-down of *nmgp-1* extends memory at 2 hr (*Figure 2h*). That is, in each of these cases, reduction of these genes did not impair memory, as loss of a compensatory function would appear; rather, loss of these age-upregulated genes improved wild-type memory. These results indicate that at least some neuronal genes that increase with age can have a negative impact on learning and memory, as demonstrated by the improvement of memory when knocked down, even in young animals. While it is still possible that some upregulated genes may act in a compensatory manner, the simplest model is that at least some are actively deleterious for learning and memory. We previously observed that for genes that play a role in complex behaviors like learning and memory, the loss of single genes can have a large impact on these complex behaviors (*Lakhina et al., 2015*), unlike the additive roles of longevity-promoting genes (*Murphy et al., 2003*). Therefore, one mechanism by which wild-type worms lose their learning and memory functions with age is not just by loss of neuronal gene expression, as one might expect, but also by dysregulation of expression of genes that can negatively impact learning and memory.

## *daf-2* mutants maintain learning and memory with age

We previously found that *daf-2* animals have extended motility (Maximum Velocity) that correlates with and predicts their extension of lifespan (*Hahm et al., 2015*). Additionally, not only do young *daf-2* worms have better memory than wild-type worms, but *daf-2* mutants also maintain learning and memory better with age (*Kauffman et al., 2010*; *Kaletsky et al., 2016*). However, the duration of this improvement was unknown. To determine the proportion of life that worms can learn and remember, we tested wild-type, *daf-2*, and *daf-16;daf-2* worms for their learning and associative memory ability every day until these functions were lost. We found that while wild-type worms lose their learning and short-term memory abilities by Day 7–8 (*Figure 1a*, *Figure 3a and b*), learning and memory span were significantly extended in *daf-2* mutants (*Figure 3a and b*); thus, a comparison of *daf-2* neurons with wild-type neurons at Day 8 should reveal differences relevant to cognitive aging. The extension of learning and memory is dependent on the FOXO transcription factor DAF-16 (*Figure 3a*); in fact, while *daf-16;daf-2* mutants still have the ability to learn for a few days, these mutants are completely unable to carry out any memory ability, even on Day 1. Thus, learning ability, which is similar in wild-type and *daf-2;daf-16* mutants, is mechanistically distinct from short-term memory ability (*Stein and*

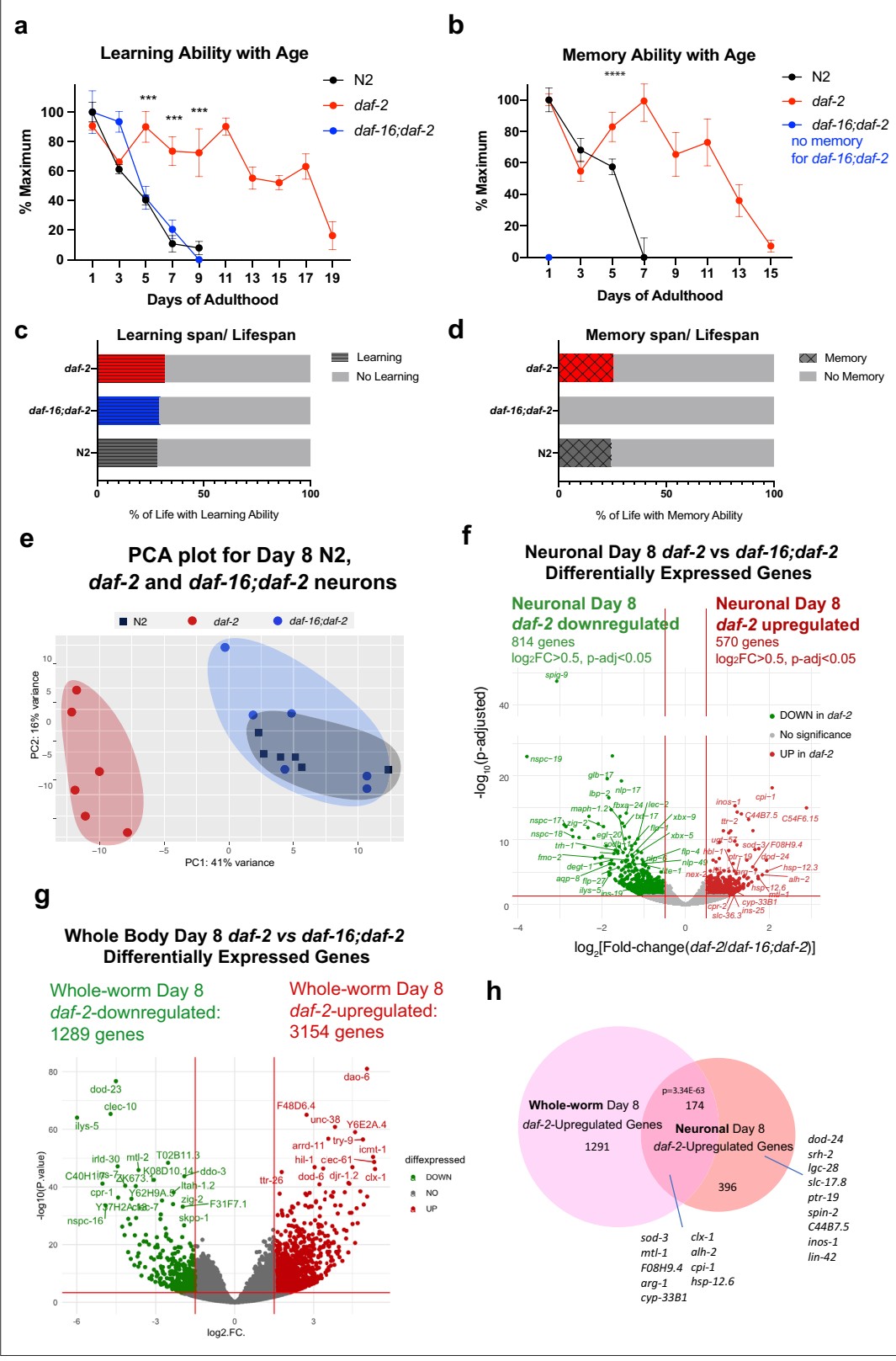

**Figure 3.** Identifying neuronal IIS/FOXO targets in aged worms using neuron-specific RNA-sequencing. (**a**) *daf-2* mutants show better learning maintenance with age compared to N2 and *daf-16;daf-2* worms. n=10 plates in each condition. (**b**) *daf-2* mutants show better memory maintenance with age compared to N2 worms. *daf-16;daf-2* worms do not have 1 hr memory on Day 1 of adulthood. N=10 plates in each condition. (**c–d**) *daf-2*

*Figure 3 continued on next page*

*Figure 3 continued*

mutants have a slightly larger learning span/lifespan ratio and memory span/lifespan ratio than N2 (wild-type). Lifespan shown in *Figure 3—figure supplement 1c*. (**e**) PCA plot of Day 8 N2, *daf-2*, and *daf-16;daf-2* neuronal RNA sequencing results. (**f**) Volcano plot of neuronal *daf-2*-regulated, *daf-16*-dependent up- and downregulated genes on adult Day 8 (Adjusted p-value < 0.05, log$_2$(Fold-change) >0.5, n=6 biological replicates per strain). 570 genes were significantly upregulated and 814 genes were downregulated in *daf-2* neurons compared with *daf-16;daf-2*. (**g**) Volcano plot of whole-worm *daf-2* vs *daf-16;daf-2* differentially-expressed genes during aging. 3154 genes are higher in *daf-2*, 1289 genes are higher in daf-16;daf-2 (log$_2$[Fold-change(*daf-2* vs *daf-16;daf-2*)]>1.5, p-adjusted <0.01). (**h**) Comparison of neuronal and whole-worm Day 8 *daf-2* differentially-expressed genes (overlap p=3.34E-63, hypergeometric test). Neuron-specific and shared *daf-2* upregulated genes with the highest fold changes are labeled.

The online version of this article includes the following figure supplement(s) for figure 3:

**Figure supplement 1.** Neuron-specific sequencing of Day 8 *daf-2* and *daf-16;daf-2* mutants.

**Figure supplement 2.** Whole-worm RNA-sequencing identifies changes in aged *daf-2* mutants.

**Figure supplement 3.** DAF-16-dependent and -independent *daf-2*-regulated genes show different features.

---

*Murphy, 2014*). *daf-2* worms maintained learning ability until Day 19 and short-term (1 hr) memory ability until Day 15, more than twice the duration of wild-type worms, while *daf-16;daf-2* worms exhibit no short-term memory ability, even on Day 1 of adulthood (*Figure 3b*). Our data suggest that the learning span-to-lifespan (*Figure 3—figure supplement 1c*) and memory span-to-lifespan ratios in *daf-2* worms were similar to or slightly higher than that of wild-type worms (*Figure 3c and d*), indicating that *daf-2* mutants maintain cognitive function for at least proportionally as long as wild-type worms do. Thus, *daf-2* mutants maintain their higher cognitive quality of life longer than wild-type worms, while *daf-16;daf-2* mutants spend their whole lives without memory ability (*Figure 3d*), in contrast to claims that *daf-2* mutants are less healthy than wild-type or *daf-16* worms (*Bansal et al., 2015*). Additionally, it should be noted that because our choice assays distinguish motility function from learning and memory function (*Kauffman et al., 2011*), the improvements in memory with age shown by *daf-2* mutants relative to wild-type are distinct from *daf-2's* improvements in motility that we previously showed (*Hahm et al., 2015*). Therefore, we are interested in these genes that might contribute to the extended cognitive function that *daf-2* worms demonstrate.

## Aging IIS/FOXO neurons express stress-resistance genes to maintain neuronal function with age

To identify genes that may improve memory and slow cognitive aging in long-lived *daf-2* mutants, we compared the transcriptional profiles of Day 8 FACS-isolated neurons from *daf-2* animals with Day 8 FACS-isolated wild-type and *daf-16;daf-2* neurons; by Day 8, wild-type and *daf-16;daf-2* worms have already lost their learning and memory ability, but *daf-2* worms still maintain their cognitive functions (*Figure 3a and b*). It should be noted that wild-type worms still have normal chemotaxis and motility at Day 8 (*Kauffman et al., 2010*), and there is a separation of several days between the loss of cognitive functions and the loss of motility (*Kauffman et al., 2010*); therefore, comparison of the neuronal transcriptomes of *daf-2* with wild-type and *daf-16;daf-2* at this age should specifically highlight genes that are required for learning and memory rather than other functions.

The PCA of the *daf-2*, *daf-16;daf-2*, and wild-type neuronal Day 8 transcriptomes (*Figure 3e*) indicates that aged *daf-16;daf-2* mutant neurons are similar to aged wild-type neurons, correlating well with their similarly worsened cognitive functions at this age; that is, at a transcriptomic level, aged (Day 8) wild-type neurons and aged *daf-16;daf-2* neurons are similar, which is echoed by their shared inability to carry out learning and memory by Day 8 of adulthood (*Figure 3ea, b*). By contrast, the transcriptomes of aged *daf-2* mutant neurons are distinct from both aged wild-type and aged *daf-16;daf-2* neuron transcriptomes, just as the cognitive abilities of *daf-2* are much greater than wild-type or *daf-16;daf-2* at this age. Downsampling analysis shows that our sequencing depth is sufficient to saturate the detectable differential expression (*Figure 3—figure supplement 1g, h*). We obtained an average of 47,233,119 counted reads per sample (*Supplementary file 5*) and detected the expression of 16,488 coding and non-coding genes (*Figure 3—figure supplement 1e*).

We identified 570 upregulated and 814 downregulated genes in Day 8 *daf-2* neurons compared to Day 8 *daf-16;daf-2* neurons (*Figure 3f*). A large fraction of the downregulated genes in Day 8 *daf-2* vs *daf-16;daf-2* neurons are 'nematode-specific peptide family' (*nspc*-) genes of unknown function (*Figure 3f*). While the *daf-2* vs *daf-16;daf-2* changes in whole worms largely replicated the results from our previous studies of young animals (*Murphy et al., 2003*; *Tepper et al., 2013*; *Figure 3g*, *Supplementary file 6g*), comparison of the *daf-2* vs *daf-16;daf-2* differential transcriptional changes in Day 8 whole worms and Day 8 neurons reveal shared (174 genes) and neuron-specific gene expression changes (396 genes; *dod-24, srh-2, lin-42*, etc.) (*Figure 3h*, *Figure 3—figure supplement 2c*). Not surprisingly, previously identified genes from whole-worm *daf-2* vs *daf-16;daf-2* and N2 (e.g. *sod-3, mtl-1, cpi-1, hsp-12.6*, etc.) that play roles in both neurons and other tissues even in Day 1 *daf-2* mutants appear in the shared list (Some neuron-specific Day 8 *daf-2*-upregulated genes have not been reported to be expressed in neurons previously (e.g. *spin-2*), further suggesting the value of transcriptomic analyses of isolated neurons in mutant backgrounds at this age).

Many genes upregulated in Day 8 *daf-2* neurons relative to *daf-16;daf-2* are related to stress responses, including heat stress (e.g. *hsp-12.6, hsp-12.3, F08H9.4/hsp*), oxidative stress (e.g. *sod-3*), and metal stress genes (e.g. *mtl-1*); and proteolysis (e.g. *cpi-1, cpr-2*, and *tep-1*). The upregulation of these genes may perform neuroprotective functions, as their homologs in mammals have been shown to do (*Table 1*). Specifically, 36 of the top 100 upregulated genes have identified orthologs or identified domains with known functions, of which 32 of (89%) have functions in promoting neuronal health. These mammalian homologs protect neurons against protein aggregation and harmful metabolites (e.g. *cpi-1, alh-2, ttr-41, gpx-5*) (*Gauthier et al., 2011*; *Carmichael et al., 2021*; *Li et al., 2011*; *Lee et al., 2020*; *Hambright et al., 2017*), maintain synaptic organization and neuronal homeostasis (e.g. *dod-24, ptr-19, plep-1*) (*González-Calvo et al., 2022*; *Ung et al., 2018*; *Perland et al., 2016*), facilitate neuronal injury repair (e.g. *F08H9.4, sod-3*) (*Huang et al., 2023*; *Flynn and Melov, 2013*), and maintain normal neuronal function (e.g. *lgc-28, slc-36.3, lin-42*) (*Koukouli and Changeux, 2020*; *Zeiger et al., 2008*; *Lautrup et al., 2019*; *Smies et al., 2022*). Together, these genes may help maintain *daf-2*'s neuronal health and protect neurons from accumulation of environmental harm during aging.

We found that about a third of the *daf-2*-upregulated genes were shared between the *daf-2* vs *daf-16;daf-2* analysis and the *daf-2* vs N2 analysis (338 genes) (*Figure 3—figure supplement 3*). Of the unshared genes, the *daf-2*-maintained genes that are specific to the *daf-2* vs N2 comparison are bZIP transcription factors, including *zip-5, zip-4, atf-2*, and proteasome components (*Figure 3—figure supplement 3D*). These results indicate that other transcription factors may participate in regulating *daf-2* functions in aged neurons in addition to the *daf-16*/FOXO transcription factor.

## IIS/FOXO transcriptomic changes are necessary for *daf-2* mutant's improved neuronal functions

We were interested not only in the genes that remained upregulated with age, but also in genes that might have increased with age in the high-performing *daf-2* mutants. That is, are there genes that increase in expression in *daf-2* mutants that are necessary or beneficial for their continued high performance with age? Some of the Day 8 *daf-2* vs wild-type or *daf-16;daf-2* upregulated genes are also Class 1 DAF-16-dependent genes (*Murphy et al., 2003*) (*sod-3, hsp-12.3, fat-5*, and *mtl-1, hil-1*, and *dao-2*). However, many more genes were differentially expressed in Day 8 *daf-2* vs *daf-16;daf-2* neurons from our Day 1 data (*Kaletsky et al., 2016*; *Figure 4a*, *Figure 3—figure supplement 2d*). Of the 'new' genes – that is, genes upregulated specifically in neurons of Day 8 vs Day 1 of *daf-2* vs *daf-16;daf-2* – many have mammalian homologs that have been shown to play neuroprotective roles, by protecting against aggregation proteins and harmful metabolites, maintaining synaptic organization, neuronal homoeostasis, or neuronal activity, or facilitating neuronal injury repair (see *Table 1* for specific references).

If the upregulated genes in aged *daf-2* neurons are responsible for the extended memory span of *daf-2* mutants, knocking down those genes should block older *daf-2* mutants' memory functions. Therefore, we tested the effect of RNAi knockdown of the top fold-change candidate genes on *daf-2*'s memory in aged adults. We chose Day 6 for testing because by then, like on Day 8, wild-type worms have already lost their learning and most memory abilities, but *daf-2* worms retain normal cognitive functions, and this time point avoids the increased naïve chemotaxis that we observe in older *daf-2*

**Table 1.** List of top *daf-2* vs *daf-16;daf-2* upregulated genes with orthologs that have neuroprotective functions.

| Gene name | Full name | log2(FC) | p-adj | Mammalian ortholog | Ortholog full name | Inferred function |
|---|---|---|---|---|---|---|
| **Neuroprotective against Neurodegenerative Diseases** | | | | | | |
| *cpi-1* | Cysteine Protease Inhibitor 1 | 2.07 | 7.80E-19 | CST3 | Cystatin C | Protease inhibitor, suppresses AD pathology **Gauthier et al., 2011** |
| *alh-2* | ALdehyde deHydrogenase 2 | 1.83 | 2.65E-05 | ALDH1A1 | Aldehyde dehydrogenase 1 | Expressed in dopaminergic neurons. Regulates dopamine release in Parkinson's Disease **Carmichael et al., 2021** |
| *ttr-41,45,2* | TransThyretin-Related family domain 41,45,2 | 1.68 | 3.98E-06 | | | Inhibits Aβ fibril formation, and suppresses the AD pathology **Li et al., 2011** |
| *cyp-33B1* | CYtochrome P450 family 33B1 | 1.34 | 2.04E-03 | CYP2J2 | Cytochrome P450 2J2 | Protective against Parkinson's Disease through altered metabolism **Li et al., 2018**; **Ferguson and Tyndale, 2011** |
| *spin-2* | SPINster (Dm lysosomal permease) homolog 2 | 1.27 | 6.20E-04 | SPNS2 | Spinster homolog 2 | Sphingosine-1-phosphate Transporter, neuroprotective in AD **Zhong et al., 2019** |
| *gpx-5* | Glutathione PeroXidase 5 | 1.27 | 3.99E-04 | GPX3,5,6 | glutathione peroxidase 3,5,6 | Protects again lipid peroxidation, protects against neurodegeneration **Lee et al., 2020**; **Hambright et al., 2017** |
| *cpr-2* | Cysteine PRotease related 2 | 1.25 | 5.01E-03 | CTSB | Cathepsin B | Lysosomal Protease, Involved in Aβ and APP protein degradation **Cermak et al., 2016** |
| *djr-1.2* | DJ-1 (mammalian transcript'l regulator) Related 1.2 | 1.09 | 3.90E-03 | PARK7 | Parkinsonism associated deglycase | Neuroprotective against Parkinson's Disease; Prevents accumulation of harmful metabolites **Heremans et al., 2022** |
| **Synaptic Organization Maintenance** | | | | | | |
| *dod-24* | Downstream Of DAF-16 (regulated by DAF-16) 24 | 1.93 | 1.39E-07 | | Cub-like Domain Containing Protein | Clustering of neurotransmitter receptor proteins **González-Calvo et al., 2022** |
| *ptr-19,15* | PaTched Related family 19,15 | 1.21 | 1.72E-05 | PTCHD1,3,4 | Patched domain-containing 1,3,4 | Synaptic organization, autism risk factor **Ung et al., 2018**; **Pastore et al., 2022** |
| *hbl-1* | HunchBack Like (fly gap gene-related) 1 | 1.16 | 6.47E-06 | hb | Hunchback (fly) | Regulate synapse number and locomotor circuit function **Lee et al., 2022** |
| *cutl-4* | CUTiclin-Like 4 | 1.08 | 2.74E-02 | pio | Piopio (fly) | ECM protein for axonal growth and synapse formation **Broadie et al., 2011** |
| *lron-2* | eLRR (extracellular Leucine-Rich Repeat) ONly 2 | 1.06 | 8.70E-05 | LGI1,2 | Leucine-Rich Glioma Inactivated protein 1 | Modulation of trans-synaptic proteins. Protection against seizure **Fels et al., 2021** |
| **Neuronal Homeostasis Maintenance** | | | | | | |
| *mocs-1* | MOlybdenum Cofactor Sulfurase 1 | 1.05 | 1.17E-04 | MOCOS | Molybdenum cofactor sulfurase | Regulation of redox homeostasis and synaptogenesis. Down in ASD **Rontani et al., 2021** |
| *plep-1* | PLugged Excretory Pore 1 | 1.12 | 2.92E-03 | MFSD11 | Major facilitator superfamily domain 11 | Putative SLC solute carrier protein, involved in brain energy homeostasis **Perland et al., 2016** |
| *cky-1* | CKY homolog 1 | 1.08 | 1.58E-04 | NPAS4 | Neuronal PAS Domain Protein 4 | Calcium-dependent transcription factor, neuronal homeostasis maintenance **Fu et al., 2020**; **Shan et al., 2018** |
| **Neuronal Injury Repair facilitation** | | | | | | |
| *F08H9.4, hsp-12.3,12.6* | small HSP domain-containing protein | 1.94 | 7.33E-06 | HSPB2 | Heat-shock Protein Beta 2 | Facilitates PNS injury regeneration, suppresses inflammation **Huang et al., 2023** |
| *sod-3* | SOD superoxide dismutase 3 | 1.66 | 3.05E-09 | SOD2 | superoxide dismutase2 | Converts superoxide to the less reactive hydrogen peroxide ($H_2O_2$). Protects neurons from injury. **Flynn and Melov, 2013** |

*Table 1 continued on next page*

*Table 1 continued*

**Normal Neuronal Activity Maintenance**

| | | | | | | |
|---|---|---|---|---|---|---|
| *lgc-28* | Ligand-Gated ion Channel 28 | 1.38 | 7.29E-04 | CHRNA6,3 | Neuronal acetylcholine receptor subunit alpha-6,3 | Nicotinic receptor. Regulates cognitive functions and addiction ***Koukouli and Changeux, 2020***; ***Zeiger et al., 2008*** |
| *F22B7.9* | | 1.33 | 8.91E-15 | METTL23 | methyltransferase like 23 | Interacts with GABPA; disruption causes intellectual disability ***Bernkopf et al., 2014***; ***Reiff et al., 2014*** |
| *fat-5* | FATty acid desaturase 5 | 1.31 | 3.40E-03 | SCD5 | StearoylCoA Desaturase-5 | Neuronal Cell Proliferation and Differentiation ***Sinner et al., 2012*** |
| *slc-36.3* | SLC (SoLute Carrier) homolog 36.3 | 1.25 | 2.88E-03 | SLC36A4 | Solute Carrier Family36 Member4 | amino acid transporter, transports Trp, involved in kynurenic acid pathway ***Lautrup et al., 2019*** |
| *lin-42* | abnormal cell LINeage 42 | 1.15 | 1.66E-04 | PER1,2 | Period 1,2 | Phosphorylates CREB, modulates CREB-mediated memory consolidation ***Smies et al., 2022*** |
| *ctsa-1.1* | CaThepSin A homolog 1.1 | 1.07 | 4.97E-05 | CTSA | Lysosomal Ser carboxy-peptidase Cathepsin A | Involved in normal neuronal development ***De Pasquale et al., 2020***; ***Hsu et al., 2018***; |
| *gsnl-1* | GelSoliN-Like 1 | 1.06 | 2.93E-04 | AVIL | advillin | Facilitates somatosensory neuron axon regeneration ***Chuang et al., 2018*** |

animals. As shown in *Figure 4b–c*, we selected these significantly differentially expressed candidate genes based on their ranking in fold-change. Previously, we have found that the top significantly differentially-expressed genes (by fold-change) are most likely to have strong effects on function, while less differentially-changed genes have less of an effect (*Murphy et al., 2003*; *Kaletsky et al., 2016*; *Lakhina et al., 2015*), therefore, we prioritized genes that are significantly different and the most highly expressed in *daf-2* mutants compared to *daf-16;daf-2* mutants for subsequent testing (*Figure 4c*). *daf-2* worms, including neurons, are susceptible to RNA interference (*Kaletsky et al., 2016*; *Wang, 2004*). Of the eight candidate genes we tested, the reduction of three of them (*F08H9.4*, *mtl-1*, and *dod-24*, originally classified as a Class II gene with proposed immune activity) significantly decreased *daf-2*'s learning ability on Day 6 (*Figure 4d*). Those genes plus reduction of two additional genes (*C44B7.5* and *alh-2*) affected 1 hr memory (*Figure 4e*) in Day 6 *daf-2* mutants. That is, knock-down of the heat shock-related gene *F08H9.4*, the innate immunity gene *dod-24*, aldehyde dehydro-genase *alh-2,* and previously uncharacterized gene *C44B7.5* are required to some degree for *daf-2*'s extended memory ability. The reduction of the metal stress gene *mtl-1*, which is expressed in neurons as well as the rest of the body, had a slight effect on learning and memory.

One caveat of these experiments is that, while we found these genes through the isolation of neurons from aged worms and subsequent RNA-sequencing, the knockdown of the genes and its effects are not necessarily neuron-autonomous; however, *alh-2* and *F08H9.4* were reported to only be expressed in neurons and the cephalic sheath cell (*Kaletsky et al., 2018*), and *C44B7.5* and *dod-24*, while expressed more broadly, were not upregulated in *daf-2* in the whole-worm analysis (*Figure 3f*), therefore, their effects are most likely neuron-autonomous. In fact, *dod-24* is one of the original Class 2 *daf-2*-downregulated genes from whole-animal analyses, suggesting that *dod-24*'s increase in expression is specifically in neurons, therefore, the effect of its knockdown is most likely to be neuron-autonomous.

Together, these data suggest that the specific genes that are differentially regulated in Day 8 *daf-2* mutants may aid in slowing neuronal function decline and behavioral changes associated with aging. Furthermore, memory maintenance with age might require additional genes that function in promoting stress resistance and neuronal resilience, which were not previously uncovered in analyses of young animals.

## Discussion

Although it has been shown previously that *daf-2* worms maintain various functions with age, how long they can maintain learning and memory with age, and the genes that might be responsible

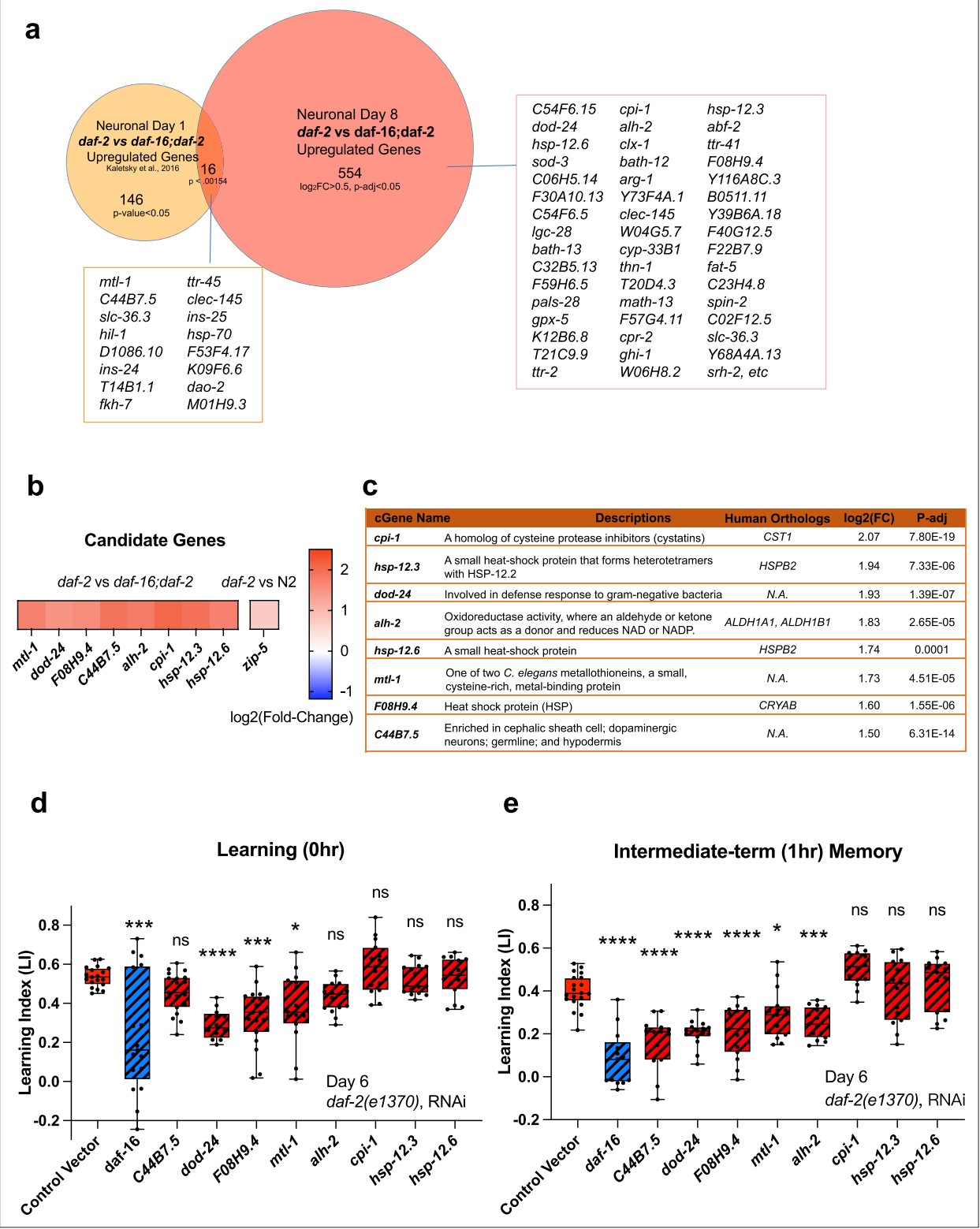

**Figure 4.** Neuronal IIS/FOXO aging targets regulate memory decline with age in *daf-2* worms. (**a**) Comparison of neuronal Day 1 and Day 8 *daf-2* vs *daf-16;daf-2* upregulated genes. All shared genes and top Day 8-specific *daf-2* upregulated genes are labeled. (**b**) *daf-2*-regulated fold-change profile of candidate genes. All candidates are upregulated in *daf-2* mutants. (**c**) Description of candidate genes. log₂(Fold-change) and p-adjusted values from the *daf-2* vs *daf-16;daf-2* comparison unless stated otherwise. (**d**) Candidate gene knockdown effects on Day 6 adult *daf-2* learning (0 hr after conditioning). Two candidate genes, *dod-24* and *F08H9.4*, show a significant decrease in learning ability. N=5 plates in each condition, merged results of 3 biological

*Figure 4 continued on next page*

*Figure 4 continued*

repeats shown. (**e**) Candidate gene knockdown effects on Day 6 adult *daf-2* short-term memory (1 hr after conditioning). *C44B7.5*, *dod-24*, *F08H9.4*, *mtl-1,* and *alh-2* showed significant decreases in memory. n=5 plates in each condition, the representative image of three biological repeats shown. (**d-e**) RNAi was performed using a neuron-sensitized RNAi strain CQ745: *daf-2(e1370) III; vIs69 [pCFJ90(Pmyo-2::mCherry +Punc-119::sid-1)] V.*\*p<0.05. \*\*p<0.01. \*\*\*p<0.001. \*\*\*\*p<0.0001. One-way ANOVA with Dunnet's post-hoc analysis. Box plots: center line, median; box range, 25-75$^{th}$ percentiles; whiskers denote minimum-maximum values.

for these extended neuronal functions, have not been previously explored. Here, we have found that *daf-2* worms maintain learning and memory abilities proportional with (or even slightly beyond) their degree of lifespan extension, underscoring *daf-2's* improved healthspan (*Hahm et al., 2015*). To understand how memory is lost with age and retained in insulin/IGF-1-like signaling mutants, we have characterized the neuronal transcriptomes of aged wild-type worms and IIS (*daf-2*) and IIS/FOXO (*daf-16;daf-2*) mutants (*Figure 5*). We found that wild-type neuronal aging is characterized by a down-regulation of neuronal function genes and an upregulation of proteolysis genes and transcriptional and epigenetic regulators, which together may help explain the loss of neuronal identity and function with age. We also identified the transcriptomic profile accompanying *daf-2's* extended learning and memory span. Specifically, *daf-2* neurons maintain higher expression of stress response genes and predicted neuronal homeostasis functions (*Table 1*), which may help make them more resistant to environmental adversities and age-related decline. We also identified genes responsible for wild-type worms' worsened learning and memory with age.

By employing a FACS-based neuron-sorting technique, we selectively analyzed adult neuron-function-related genes and investigated their aging process, which is not easily discernible through

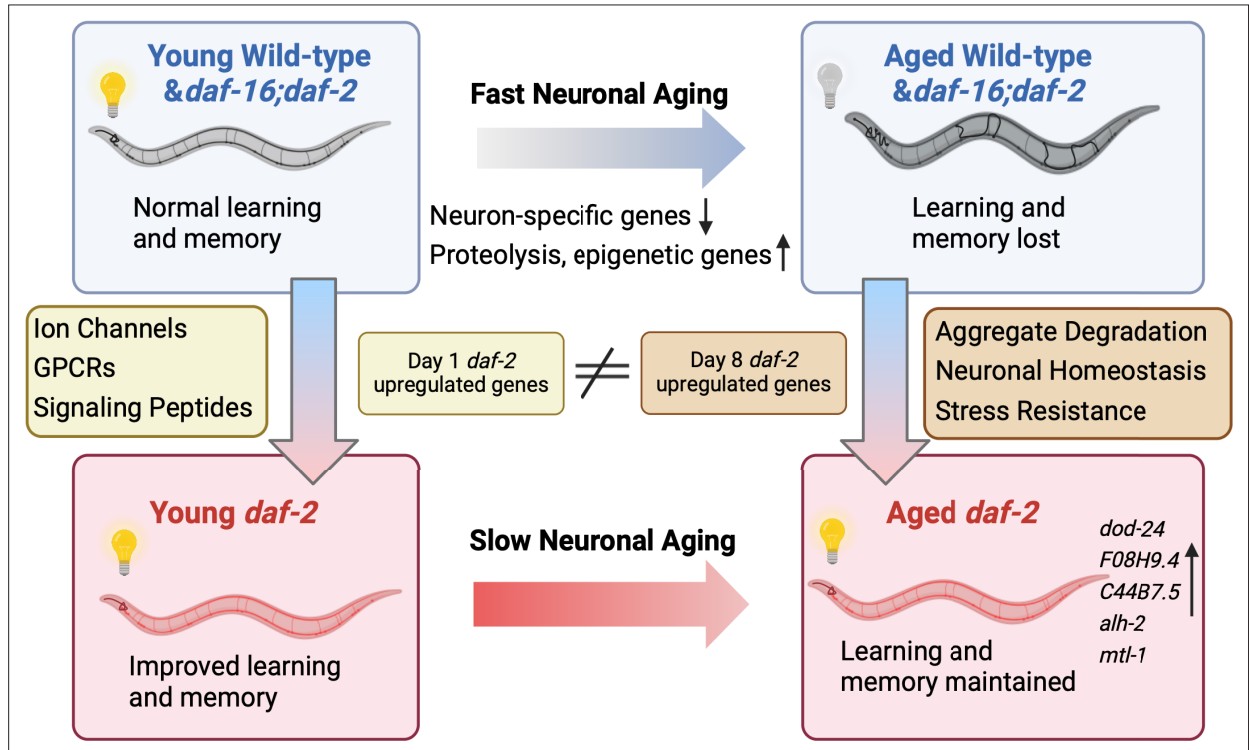

**Figure 5.** Aged *daf-2* neurons upregulate neuroprotective genes to maintain improved cognitive behaviors. During normal neuronal aging, neuron-specific genes decrease in expression, while proteolysis and epigenetic regulators are upregulated, resulting in neuron dysfunction and cognitive function loss. In aged *daf-2* mutants, upregulation of neuroprotective genes including *dod-24*, *F08H9.4*, *C44B7.5*, *alh-2*, and *mtl-1* contribute to *daf-2's* improved cognitive function. The diagram was generated using Biorender, and published using a CC BY-NC-ND license with permission.

The online version of this article includes the following figure supplement(s) for figure 5:

**Figure supplement 1.** Comparison with recent sequencing datasets.

whole-worm sequencing (*Gao et al., 2023*; *Wang et al., 2022*; *Roux et al., 2023*; *Figure 5—figure supplement 1*). Sequencing many biological repeats of aging neurons to high depth with ribosomal RNA depletion allowed us to detect a larger number of genes compared to other neuron-related bulk and single-cell sequencing profiles (*Wang et al., 2022*), providing a deep transcriptomic dataset of aged wild-type, IIS mutant, and IIS/FOXO mutant neurons. Our analysis allowed us to identify differentially expressed genes that are known to be expressed in at a small number of neurons, even for low-abundance genes. Notably, our sequencing results uncovered genes previously not known to be expressed in neurons that remained undetected in other datasets. Moreover, we revealed the involvement of known neuronal genes in the aging process, such as *ins-6* and *srd-23*. We hope that this dataset will become a valuable resource for detecting new candidates in neuronal aging.

For example, *dod-24*, which we observed to be upregulated in *daf-2* neurons and required for *daf-2*'s extended memory, was downregulated in the *daf-2* whole-worm transcriptome (*Figure 3—figure supplement 2c*). *dod-24* has been traditionally classified as a Class II gene that is downregulated in *daf-2* worms and upregulated by *daf-16* RNAi treatment (*Murphy et al., 2003*; *Tepper et al., 2013*). Functionally, it has been shown to be an innate immunity gene upregulated during pathogen infection (*Shapira et al., 2006*; *Eckl et al., 2017*; *Mack et al., 2022*), and its whole-body reduction has been shown to extend the lifespan of wild-type animals (*Murphy et al., 2003*). However, here we find that *dod-24* is beneficial in the nervous system and required for *daf-2*'s extended learning and memory in aged worms. This intriguing contrast between the whole-worm transcriptome and the neuron-specific transcriptome suggests that some genes may have distinct regulatory roles in the nervous system, necessitating a more precise approach beyond whole-worm transcriptomics.

Using this neuron-specific sequencing profile of aged cells, we identified key pathways that change during neuron aging. Our sequencing of aged neurons uncovered active transcriptomic alterations during aging, resulting in not just transcriptional silencing but also upregulation of various pathways. We found that the Day 8 *daf-2* vs *daf-16;daf-2* neuronal differentially-expressed genes that we have newly discovered here differ from the neuronal Day 1 *daf-2* vs *daf-16;daf-2* dataset we previously obtained (*Kaletsky et al., 2016*). These Day 8 differentially-expressed genes are not canonical neuronal genes, such as receptors and ion channels; instead, there are more metabolic and proteolytic genes whose protein orthologs have been shown to be neuroprotective. All top 50 genes and 90% of the top 100 genes with identified mammalian orthologs have been shown to be essential to neuronal functions in mammals (*Table 1*). These results indicate that instead of simply mimicking a young state, *daf-2* may enhance neuron's resilience to the accumulation of harm and take a more active approach to combat aging. These changes suggest that *daf-2*'s extended memory maintenance may require different mechanisms than function in young animals; *daf-2* may maintain neuronal function not just by retaining a youthful transcriptome, but also by increasing the expression of genes that promote resilience, such as stress-response genes and proteolysis inhibitors.

In addition to examining aging in wild-type and IIS/FOXO mutants independently, our results further linked the normal aging process to altered gene regulation in the IIS pathway. *utx-1*, *nmgp-1*, and *ins-19* increase in expression in aged neurons, and we found that their reduction improved memory, indicating that at least some of the genes whose expression rises with age can have a negative impact on normal cognitive functions, rather than acting in a compensatory manner. *utx-1* is an H3K27me3 histone demethylase we found to be higher in wild-type aged neurons, but it is also involved in the IIS pathway. The downregulation of *utx-1* has been shown to regulate development (*Vandamme et al., 2012*) and promote longevity (*Jin et al., 2011*; *Maures et al., 2011*; *Guillermo et al., 2021*), and its mammalian homolog has been implicated in regulating cognitive abilities (*Shaw et al., 2023*; *Tang et al., 2017*). The longevity response of *utx-1* depends on *daf-16* (*Jin et al., 2011*; *Maures et al., 2011*; *Guillermo et al., 2021*). The loss of *utx-1* decreases methylation on the *daf-2* gene, thus increasing DAF-16's nuclear localization, mimicking a *daf-2* mutation (*Jin et al., 2011*). This example of the crosstalk between normal aging and IIS/FOXO mutants offers valuable insights into modifying the aging process for enhanced longevity and cognitive health.

We found that the insulin-like peptide *ins-19* was upregulated in aged neurons and was downregulated in aged *daf-2* neurons, and its downregulation in wild-type worms extended memory span. Insulin-like peptides play crucial roles as receptor ligands (in both agonist and antagonist roles) for DAF-2, and we have found them to be downregulated in *daf-2* mutants compared with *daf-16;daf-2* mutants, possibly creating a feedback loop that dampens the insulin signaling pathway,

as was previously shown for *ins-7* and *ins-18* (*Murphy et al., 2003*; *Murphy et al., 2007*). These peptides exhibit diverse functions in development, dauer formation, and longevity (*Murphy et al., 2003*; *Murphy et al., 2007*; *Thomé-Duret et al., 1998*; *Kawano et al., 2000*; *Pierce et al., 2001*; *Li et al., 2003*). Notably, certain insulin-like peptides have been linked to neuronal activities, such as the regulation of aversive learning by the two antagonistic peptides *ins-6* and *ins-7* (*Chen et al., 2013*), and reduced long-term learning and memory by *ins-22* RNAi (*Lakhina et al., 2015*). In our study, the expression changes of *ins-19* during wild-type aging and in *daf-2* mutants provide an example of how longevity mutants can reverse wild-type transcriptional changes during aging, ultimately reducing behavioral and functional decline.

## Summary of mechanistic insights

Our analysis of transcriptomes from isolated aged Day 8 neurons of wild-type, *daf-2,* and *daf-16;daf-16* mutants reveal several major mechanistic insights. Specifically, we found that wild-type neurons lose their neuronal identity through a combination of the loss of neuron-specific function genes with age, and the concomitant dysregulated increase in non-neuronal genes with age. Furthermore, at least a fraction of the top-upregulated genes with age can play deleterious roles; that is, they rise with age, and their knockdown improves function, even in young animals. This argues against the idea that all of these genes play a compensatory role with age.

We also found that the knockdown of individual top-ranked genes that function in learning and memory can have a large impact - like removing a cog of a machine. This is in contrast to our earlier findings regarding gene reduction in lifespan, where most cellular longevity processes regulated by DAF-16 activity appear to be additive, and therefore loss of individual major genes downstream of DAF-2 and DAF-16 have at most a 5–10% impact (*White et al., 1986*; *Murphy et al., 2003*). Several of these genes we found to be required for *daf-2*'s age-related improvement in learning and memory - namely *dod-24*, *F08H9.4*, *C44B7.5*, and *alh-2* – were previously not associated with memory function. Finally, these genes are distinct from the set of upregulated Day 1 *daf-2* vs *daf-16;daf-2* genes; how they each individually maintain neuronal function better with age will be interesting to dissect.

## Conclusions

Beyond our sequencing analysis, we have established links between genomics, function, and behavior. We also identified several new genes required for *daf-2*'s age-related improvement in learning and memory, shedding light on their neuron-specific roles. These additional findings further suggest that neuronal sequencing datasets can be used to identify functional candidates and pathways during the aging process. By bridging the gap between transcriptomic landscapes, genetic regulation, and functional outcomes, our study provides a greater understanding of the mechanisms underlying neuronal aging, providing insights into the development of aging interventions.

## Methods

### Strains and worm cultivation

N2 (wild-type), OH441: otIs45(*unc-119::GFP*), CQ295: otIs45(*unc-119::GFP*);*daf-2(e1370)*, CQ296: otIs45(*unc-119::GFP*);*daf-16(mu86);daf-2(e1370)*), LC108: uIs69 (*myo-2p::mCherry +unc-119p::sid-1*), CQ705: daf-2(e1370) III, 3 X outcrossed, CQ745: *daf-2(e1370) III;* vIs69 [*pCFJ90(Pmyo-2::mCherry +Punc-119::sid-1)*] *V*, QL188: *ins-19(tm5155) II*, CX3695: kyIs140(*str-2::GFP +lin-15(+)*), CQ461*: (daf-2(e1370);Pmec-4::mCherry)*, and CQ501*: (daf-2 (e1370);zip-5(gk646);Pmec-4::mCherry)*. Strains were grown on high-growth media (HGM) plates seeded with *E. coli* OP50 bacteria using standard methods *Brenner, 1974*.

### Tissue-specific isolation

For neuronal isolation, five plates of fully-grown worms from HG plates were synchronized by hypochlorite treatment, eggs spread on seeded plates to hatch, and at least five plates/replicate were grown to L4 on HGM plates until transferred to HGM plates with FUdR to avoid progeny contamination. This gives us ~6000 healthy Day 8 worms to sort. Neuron isolation and Fluorescent-activated cell sorting were carried out as previously described (*Kaletsky et al., 2016*; *Kaletsky et al., 2018*). Briefly, worms were treated with 1000 uL lysis buffer (200 mM DTT, 0.25% SDS, 20 mM HEPES pH

8.0, 3% sucrose) for 6.5 min to break the cuticle. Then worms were washed and resuspended in 500 uL 20 mg/mL pronase from *Streptomyces griseus* (Sigma-Aldrich). Worms were incubated at room temperature with mechanical disruption by pipetting until no whole-worm bodies were seen, and then ice-cold osmolarity-adjusted L-15 buffer(Gibco) with 2% Fetal Bovine Serum (Gibco) were added to stop the reaction. Prior to sorting, cell suspensions were filtered using a 5 um filter and sorted using a FACSVantage SE w/ DiVa (BD Biosciences; 488 nm excitation, 530/30 nm bandpass filter for GFP detection). Sorting gates were determined by comparing with age-matched, genotype-matched non-fluorescent cell suspension samples. Fluorescent neuron cells were directly sorted into Trizol LS. 100,000 GFP + cells were collected for each sample.

## RNA extraction, library generation, and sequencing

We used the standard trizol-chloroform-isopropanol method to extract RNA, then performed RNA cleanup using RNeasy MinElute Cleanup Kit (Qiagen). RNA quality was assessed using the Agilent Bioanalyzer RNA Pico chip, and bioanalyzer RIN >6.0 samples were observed before library generation. 2 ng of RNA was used for library generation using Ovation SoLo RNA-Seq library preparation kit with AnyDeplete Probe Mix- *C. elegans* (Tecan Genomics) according to the manufacturer's instructions (*Barrett et al., 2021*). Library quality and concentration was assessed using an Agilent Bioanalyzer DNA 12000 chip. Samples were multiplexed and sequencing were performed using NovaSeq S1 100nt Flowcell v1.5 (Illumina).

## Data processing

FastQC was performed on each sample for quality control analysis. RNA STAR package was used for mapping paired-end reads to the *C. elegans* genome ce11 (UCSC Feb 2013) using the gene model ws245genes.gtf. Length of the genomic sequence around annotated junctions is chosen as read length –1. 50–70% of reads were uniquely mapped. Reads uniquely mapped to the genome were then counted using htseq-count (mode = union). DESeq2 analysis was then used for read normalization and differential expression analysis on counted reads (*Love et al., 2014*). Genes with a $\log_{10}$TPM >0.5 were considered as detected and genes with a $\log_2$(fold-change) >0.5 and p-adjusted <0.05 are considered differentially expressed in further analysis. Gene ontology analysis were performed using gprofiler (*Raudvere et al., 2019*) or WormCat 2.0 (*Holdorf et al., 2020*) and category 2 was selected to show. Tissue query was performed on the top 500 highest fold-change genes, using the worm tissue query website (https://www.worm.princeton.edu; *Kaletsky et al., 2018*), and only major systems were selected in the analysis.

## Learning and memory experiments

We performed Short-Term Associative Memory (STAM) experiments as previously described (*Kauffman et al., 2010*; *Kauffman et al., 2011*). Briefly, we used five plates of synchronized adult worms/samples to perform each experiment. One plate of worms was used to test the naïve chemotaxis assay without conditioning, while the other three plates were washed into M9, and washed three additional times to get rid of the bacteria. These worms are starved for 1 hr to prime them for food uptake. Then these worms are transferred to conditioning plates with NGM plates seeded with OP50 and 10% butanone stripes on the lid for 1 hr to perform conditioning. After conditioning, worms are either transferred from the conditioning plate directly to the chemotaxis plates to assess learning, or transferred to the holding plate for 1 hr or 2 hr to assess for memory. After staying on holding plates for 1 hr or 2 hr, worms are then transferred onto chemotaxis plates to assess for short-term memory. Chemotaxis assays were performed by transferring worms onto chemotaxis plates with 1 uL 10% butanone and 1 uL ethanol spots separated by 8 cm on a 10 cm NGM plate. Worms who have reached either the butanone spot or the ethanol spot are paralyzed by the 1 uL 7.5% NaN3 on these spots. For each timepoint, five chemotaxis plates are used to minimize the variation of the outcome. We performed this chemotaxis assay to butanone on naïve and appetitive-trained worms at different time points to assess change in preference to butanone.

Chemotaxis index is calculated as (# of worms at butanone-# of worms at ethanol)/(total # of worms - # of worms at origin).

Learning index is calculated by subtracting the trained chemotaxis index with naïve chemotaxis index.

For learning and memory span assays, we obtained synchronized worms from hypochlorite-treated eggs. Synchronized worms were washed onto 5'-fluorodeoxyuridine (FUdR) at L4 and maintained on FUdR plates by transferring to new plates every 2 days. 1 Day Prior to experiments, worms are washed onto fresh HG plates without FUdR to avoid change in behavior caused by FUdR. To verify that FUdR has no effect on short-term memory, we compared worms with and without FUdR (*Figure 5—figure supplement 1d*), and found no differences. For *utx-1* and *nmgp-1* RNAi experiments, synchronized L4 neuron-RNAi sensitized worms were washed onto HGM plates with carbenicillin and IPTG and seeded with HT115 RNAi bacteria containing the RNAi constructs from the Ahringer Library. For *daf-2* upregulated candidates' RNAi experiments, synchronized L4 *daf-2* neuron-RNAi-sensitized worms were washed onto HGM plates added with carbenicillin, FUdR, and isopropyl-b-D-thiogalactopyranoside (IPTG) and seeded with HT115 bacteria containing RNAi constructs generated from the Ahringer RNAi Library, then were transferred onto fresh RNAi plates every 2 days until Day 6. 1 Day Prior to experiments, worms are transferred onto plates without FUdR.

## Quantitative and statistical analysis

All experimental analysis was performed using Prism 8 software. Two-way ANOVA with Tukey post-hoc tests were used to compare the learning curve between control and experimental groups. One-way ANOVA followed by Dunnet post-hoc tests for multiple comparisons was performed to compare learning or 2 hr memory between various treatment groups and control. Chi-square test was performed to compare the neuron morphology change between young and aged AWC neurons. All GO term analyses were performed using Wormcat 2.0 software with Bonferroni corrected adjusted p-values. Venn diagram overlaps were compared using the hypergeometric test. Differential expression analysis of RNA-seq were performed using DESeq2 algorithm and adjusted p-values were generated with Wald test using Benjamini and Hochberg method (BH-adjusted p-values). Additional statistical details of experiments, including sample size (with n representing the number of chemotaxis assays performed for behavior, RNA collections for RNA-seq, and the number of worms for microscopy), can be found in the methods and figure legends. Regression analyses were performed using sklearn packages. Correlations were calculated using the SciPy packages.

## Materials availability

Further information and requests for resources and reagents should be directed to and will be fulfilled by Coleen T. Murphy (ctmurphy@princeton.edu).

## Acknowledgements

We thank the *Caenorhabditis* Genetics Center (CGC) for strains, WormBase (version WS289) for information, Jasmine Ashraf, William Keyes, Yichen Weng, and Titas Sengupta for help with the experiments, R Arey and other Murphy lab members for an early replicate of *daf-2* learning and memory with age, members of the Murphy Lab for input on the manuscript, Christina DeCoste, Katherine Rittenbach and the Flow Cytometry Facility for cell sorting assistance, Wei Wang and the Genomics Facility for sequencing assistance, Lance Parsons, Bruce Wang, and Chen Dan for insights on data analysis, and Biorender.com for schematic design. CTM. is the Director of the Simons Collaboration on Plasticity in the Aging Brain (SCPAB), which supported the work, and the Glenn Center for Aging Research at Princeton. YW and SZ are supported by the China Scholarship Council (CSC). KM is supported by the HHMI Gilliam Fellows Program.

## Additional information

### Funding

| Funder | Grant reference number | Author |
| --- | --- | --- |
| Simons Foundation | Simons Collaboration on Plasticity and the Aging Brain | Coleen T Murphy |

| Funder | Grant reference number | Author |
|---|---|---|
| Howard Hughes Medical Institute | Gilliam Fellows Program | Katherine Morillo |
| China Scholarship Council | | Yifei Weng Shiyi Zhou |

The funders had no role in study design, data collection and interpretation, or the decision to submit the work for publication.

## Author contributions

Yifei Weng, Conceptualization, Formal analysis, Investigation, Visualization, Methodology, Writing - original draft, Writing - review and editing; Shiyi Zhou, Conceptualization, Formal analysis, Investigation, Visualization, Methodology; Katherine Morillo, Investigation; Rachel Kaletsky, Conceptualization, Methodology; Sarah Lin, Formal analysis; Coleen T Murphy, Conceptualization, Supervision, Funding acquisition, Writing - original draft, Writing - review and editing

### Author ORCIDs

Yifei Weng http://orcid.org/0009-0003-6229-8631
Rachel Kaletsky http://orcid.org/0000-0002-7176-9154
Coleen T Murphy http://orcid.org/0000-0002-8257-984X

Reviewer #1 (Public review): https://doi.org/10.7554/eLife.95621.4.sa1
Reviewer #2 (Public review): https://doi.org/10.7554/eLife.95621.4.sa2
Reviewer #3 (Public review): https://doi.org/10.7554/eLife.95621.4.sa3
Author response https://doi.org/10.7554/eLife.95621.4.sa4

# Additional files

### Supplementary files

- Supplementary file 1. Whole-worm DEseq2 results.
- Supplementary file 2. Neuronal WT Day 1 vs Day 8 DEseq2.
- Supplementary file 3. Neuronal Day 8 *daf-2* vs *daf-16;daf-2* DEseq2.
- Supplementary file 4. Neuronal Day 8 *daf-2* vs N2 DEseq2.
- Supplementary file 5. Neuronal Day 8 *daf-16;daf-2* vs N2 DEseq2.
- Supplementary file 6. Number of sequencing reads.
- Supplementary file 7. Raw behavioral data.
- MDAR checklist

### Data availability

Sequencing reads are deposited at NCBI BioProject under accession number PRJNA999305.

The following dataset was generated:

| Author(s) | Year | Dataset title | Dataset URL | Database and Identifier |
|---|---|---|---|---|
| Weng Y, Zhou S, Murphy CT | 2024 | Analysis of the Neuron-specific IIS/FOXO Transcriptome in Aged Animals Reveals Regulators of Neuronal and Cognitive Aging | https://www.ncbi.nlm.nih.gov/bioproject/?term=PRJNA999305 | NCBI BioProject, PRJNA999305 |

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
