## [Editor Report · eLife assessment]

This **fundamental** study investigates the transcriptional changes in neurons that underlie loss of learning and memory with age in *C. elegans*, and how cognition is maintained in insulin/IGF-1-like signaling mutants. The presented evidence is **compelling**, utilizing a cutting-edge method to isolate neurons from worms for genomics that is clearly conveyed with a rigorous experimental approach. Overall, this study supports that older daf-2 worms maintain cognitive function via mechanisms that are unique from younger wild type worms, which will be of great interest to neuroscientists and researchers studying ageing.

---

## [Referee Report · Reviewer #1 (Public review)]

The authors perform RNA-seq on FACS isolated neurons from adult worms at days 1 and 8 of adulthood to profile the gene expression changes that occur with cognitive decline. Supporting data are included indicating that by day 7 of adulthood, learning and memory are reduced, indicating that this timepoint or after represents cognitively aged worms. Neuronal identity genes are reduced in expression within the cognitively aged worms, whereas genes involved in proteostasis, transcription/chromatin, and the stress response are elevated. A number of specific examples are provided, representing markers of specific neuronal subtypes, and correlating expression changes to the erosion of particular functions (e.g. motor neurons, chemosensory neurons, aversive learning neurons, etc).

To investigate whether upregulation of genes in neurons with age is compensatory or deleterious, the authors reduced expression of a set of three significantly upregulated genes and performed behavioral assays in young adults. In each case, reduction of expression improved memory, consistent with a model in which age-associated increases impair neuronal function.

The authors then characterize learning and memory in wild type, daf-2, and daf-2/daf-16 worms with age and find that daf-2 worms have an extended ability to learn for approximately 10 days longer that wild types. This was daf-16 dependent. Memory was extended in daf-2 as well, and strikingly, daf-2;daf-16 had no short term memory even at day 1. Transcriptomic analysis of FACS-sorted neurons was performed on the three groups at day 8. The authors focus their analysis on daf-2 vs. daf-2;daf-16 and present evidence that daf-2 neurons express a stress-resistance gene program. They also find small differences between the N2 and daf-2;daf-16 neurons, which correlate with the observed behavioral differences, though these differences are modest.

The authors tested eight candidate genes that were more highly expressed in daf-2 neurons vs. daf-2;daf-16 and showed that reduction of 2 and 5 of these genes impaired learning and memory, respectively, in daf-2 worms. This finding implicates specific neuronal transcriptional targets of IIS in maintaining cognitive ability in daf-2 with age, which, importantly, are distinct from those in young wild type worms.

Overall, this is a strong study with rigorously performed experiments. The authors achieved their aim of identifying transcriptional changes in neurons that underlie loss of learning and memory in *C. elegans*, and how cognition is maintained in insulin/IGF-1-like signaling mutants.

---

## [Referee Report · Reviewer #2 (Public review)]

Weng et al. perform a comprehensive study of gene expression changes in young and old animals, in wild-type and daf-2 insulin receptor mutants, in the whole animal and specifically in the nervous system. Using this data, they identify gene families that are correlated with neuronal ageing, as well as a distinct set of genes that are upregulated in neurons of aged daf-2 mutants. This is particularly interesting as daf-2 mutants show both extended lifespan and healthier neurons in aged animals, reflected by better learning/memory in older animals compared with wild-type controls. Indeed, knockdown of several of these upregulated genes resulted in poorer learning and memory. In addition, the authors showed that several genes upregulated during ageing in wild-type neurons also contribute to learning and memory; specifically, knockdown of these genes in young animals resulted in improved memory. This indicates that (at least in this small number of cases), genes that show increased transcript levels with age in the nervous system somehow suppress memory, potentially by having damaging effects on neuronal health.

Finally, from a resource perspective, the neuronal transcriptome provided here will be very useful for *C. elegans* researchers as it adds to other existing datasets by providing the transcriptome of older animals (animals at day 8 of adulthood) and demonstrating the benefits of performing tissue-specific RNAseq instead of whole-animal sequencing.

The work presented here is of high quality and the authors present convincing evidence supporting their conclusions.

---

## [Referee Report · Reviewer #3 (Public review)]

Summary

In this manuscript, Weng et al. identify the neuron specific transcriptome that impacts age dependent cognitive decline. The authors design a pipeline to profile neurons from wild type and long-lived insulin receptor/IGF-1 mutants using timepoints when memory functions are declining. They discover signatures unique to neurons which validates their approach. The authors identify that genes related to neuronal identity are lost with age in wild type worms. For example, old neurons reduce the expression of genes linked to synaptic function and neuropeptide signaling and increase the expression of chromatin regulators, insulin peptides and glycoproteins. Depletion of selected genes which are upregulated in old neurons (utx-1, ins-19 and nmgp-1) leads to improved short memory function. This indicates that some genes that increase with age have detrimental effects on learning and memory. The pipeline is then used to test neuronal profiles of long-lived insulin/IGF-1 daf-2 mutants. Genes related to stress response pathways are upregulated in long lived daf-2 mutants (e.g. dod-24, F08H9.4) and those genes are required for improved neuron function.

Strengths

The manuscript is well written, and the experiments are well described. The authors take great care to explain their reasoning for performing experiments in a specific way and guide the reader through the interpretation of the results, which makes this manuscript an enjoyable and interesting read. The authors discover novel regulators of learning and memory using neuron-specific transcriptomic analysis in aged animals, which underlines the importance of cell specific deep sequencing. The timepoints of the transcriptomic profiling are elegantly chosen, as they coincide with the loss of memory and can be used to specifically reveal gene expression profiles related to neuron function. The authors discuss on the dod-24 example how powerful this approach is. In daf-2 mutants whole-body dod-24 expression differs from neuron specific profiles, which underlines the importance of precise cell specific approaches. This dataset provides a very useful resource for the *C. elegans* and aging community as it complements existing datasets with additional time points and neuron specific deep profiling.

---

## [Author Response]

The following is the authors’ response to the previous reviews.

**eLife assessment**
This fundamental study investigates the transcriptional changes in neurons that underlie loss of learning and memory with age in *C. elegans*, and how cognition is maintained in insulin/IGF-1-like signaling mutants. The presented evidence is compelling, utilizing a cutting-edge method to isolate neurons from worms for genomics that is clearly conveyed with a rigorous experimental approach. Overall, this study supports that older daf-2 worms maintain cognitive function via mechanisms that are unique from younger wild type worms, which will be of great interest to neuroscientists and researchers studying ageing.
**Public Reviews:**

**Reviewer #1 (Public Review):**
The authors perform RNA-seq on FACS isolated neurons from adult worms at days 1 and 8 of adulthood to profile the gene expression changes that occur with cognitive decline. Supporting data are included indicating that by day 7 of adulthood, learning and memory are reduced, indicating that this timepoint or after represents cognitively aged worms. Neuronal identity genes are reduced in expression within the cognitively aged worms, whereas genes involved in proteostasis, transcription/chromatin, and the stress response are elevated. A number of specific examples are provided, representing markers of specific neuronal subtypes, and correlating expression changes to the erosion of particular functions (e.g. motor neurons, chemosensory neurons, aversive learning neurons, etc).To investigate whether upregulation of genes in neurons with age is compensatory or deleterious, the authors reduced expression of a set of three significantly upregulated genes and performed behavioral assays in young adults. In each case, reduction of expression improved memory, consistent with a model in which age-associated increases impair neuronal function.The authors then characterize learning and memory in wild type, daf-2, and daf-2/daf-16 worms with age and find that daf-2 worms have an extended ability to learn for approximately 10 days longer that wild types. This was daf-16 dependent. Memory was extended in daf-2 as well, and strikingly, daf-2;daf-16 had no short term memory even at day 1. Transcriptomic analysis of FACS-sorted neurons was performed on the three groups at day 8. The authors focus their analysis on daf-2 vs. daf-2;daf-16 and present evidence that daf-2 neurons express a stress-resistance gene program. They also find small differences between the N2 and daf-2;daf-16 neurons, which correlate with the observed behavioral differences, though these differences are modest.The authors tested eight candidate genes that were more highly expressed in daf-2 neurons vs. daf-2;daf-16 and showed that reduction of 2 and 5 of these genes impaired learning and memory, respectively, in daf-2 worms. This finding implicates specific neuronal transcriptional targets of IIS in maintaining cognitive ability in daf-2 with age, which, importantly, are distinct from those in young wild type worms.Overall, this is a strong study with rigorously performed experiments. The authors achieved their aim of identifying transcriptional changes in neurons that underlie loss of learning and memory in *C. elegans*, and how cognition is maintained in insulin/IGF-1-like signaling mutants.

We thank you for the evaluation and response.

**Reviewer #2 (Public Review):**
Weng et al. perform a comprehensive study of gene expression changes in young and old animals, in wild-type and daf-2 insulin receptor mutants, in the whole animal and specifically in the nervous system. Using this data, they identify gene families that are correlated with neuronal ageing, as well as a distinct set of genes that are upregulated in neurons of aged daf-2 mutants. This is particularly interesting as daf-2 mutants show both extended lifespan and healthier neurons in aged animals, reflected by better learning/memory in older animals compared with wild-type controls. Indeed, knockdown of several of these upregulated genes resulted in poorer learning and memory. In addition, the authors showed that several genes upregulated during ageing in wild-type neurons also contribute to learning and memory; specifically, knockdown of these genes in young animals resulted in improved memory. This indicates that (at least in this small number of cases), genes that show increased transcript levels with age in the nervous system somehow suppress memory, potentially by having damaging effects on neuronal health.Finally, from a resource perspective, the neuronal transcriptome provided here will be very useful for *C. elegans* researchers as it adds to other existing datasets by providing the transcriptome of older animals (animals at day 8 of adulthood) and demonstrating the benefits of performing tissue-specific RNAseq instead of whole-animal sequencing.The work presented here is of high quality and the authors present convincing evidence supporting their conclusions. I only have a few comments/suggestions:(1) Do the genes identified to decrease learning/memory capacity in daf-2 animals (Figure 4d/e) also impact neuronal health? daf-2 mutant worms show delayed onset of age-related changes to neuron structure (Tank et al., 2011, J Neurosci). Does knockdown of the genes shown to affect learning also affect neuron structure during ageing, potentially one mechanism through which they modulate learning/memory?(2) The learning and memory assay data presented in this study uses the butanone olfactory learning paradigm, which is well established by the same group. Have the authors tried other learning assays when testing for learning/memory changes after knockdown of candidate genes? Depending on the expression pattern of these genes, they may have more or less of an effect on olfactory learning versus for e.g. gustatory or mechanosensory-based learning.(3) A comment on the 'compensatory vs dysregulatory' model as stated by the authors on page 7 - I understand that this model presents the two main options, but perhaps this is slightly too simplistic: gene expression that rises during ageing may be detrimental for memory (=dysregulatory), but at the same time may also be beneficial other physiological roles in other tissues (=compensatory).

Thank you for your original suggestions; we addressed them in the previous version of response to the reviewers.

Comments on revised version:I am satisfied with how the authors have addressed all my comments/suggestions.

Thank you for your response!

**Reviewer #3 (Public Review):**
SummaryIn this manuscript, Weng et al. identify the neuron specific transcriptome that impacts age dependent cognitive decline. The authors design a pipeline to profile neurons from wild type and long-lived insulin receptor/IGF-1 mutants using timepoints when memory functions are declining. They discover signatures unique to neurons which validates their approach. The authors identify that genes related to neuronal identity are lost with age in wild type worms. For example, old neurons reduce the expression of genes linked to synaptic function and neuropeptide signaling and increase the expression of chromatin regulators, insulin peptides and glycoproteins. Depletion of selected genes which are upregulated in old neurons (utx-1, ins-19 and nmgp-1) leads to improved short memory function. This indicates that some genes that increase with age have detrimental effects on learning and memory. The pipeline is then used to test neuronal profiles of long-lived insulin/IGF-1 daf-2 mutants. Genes related to stress response pathways are upregulated in long lived daf-2 mutants (e.g. dod-24, F08H9.4) and those genes are required for improved neuron function.StrengthsThe manuscript is well written, and the experiments are well described. The authors take great care to explain their reasoning for performing experiments in a specific way and guide the reader through the interpretation of the results, which makes this manuscript an enjoyable and interesting read. The authors discover novel regulators of learning and memory using neuron-specific transcriptomic analysis in aged animals, which underlines the importance of cell specific deep sequencing. The timepoints of the transcriptomic profiling are elegantly chosen, as they coincide with the loss of memory and can be used to specifically reveal gene expression profiles related to neuron function. The authors discuss on the dod-24 example how powerful this approach is. In daf-2 mutants whole-body dod-24 expression differs from neuron specific profiles, which underlines the importance of precise cell specific approaches. This dataset will provide a very useful resource for the *C. elegans* and aging community as it complements existing datasets with additional time points and neuron specific deep profiling.WeaknessThis study nicely describes the neuron specific profiles of aged long-lived daf-2 mutants. Selected neuronal genes that were upregulated in daf-2 mutants (e.g. F08H9.4, mtl-1, dod-24, alh-2, C44B7.5) decreased learning/memory when knocked down. However, the knock down of these genes was not specific to neurons. The authors use a neuron-sensitive RNAi strain to address this concern and acknowledge this caveat in the text. While it is likely that selected candidates act only in neurons it is possible that other tissues participate as well.

Thank you for pointing this caveat out. We have mentioned it in the figure legend.